# Demographic and Disease-Related Predictors of Socioemotional Development in Children with Neurofibromatosis Type 1 and Plexiform Neurofibromas: An Exploratory Study

**DOI:** 10.3390/cancers14235956

**Published:** 2022-12-01

**Authors:** Yang Hou, Xian Wu, Dan Liu, Staci Martin, Mary Anne Toledo-Tamula, Taryn Allen, Andrea Baldwin, Andy Gillespie, Anne Goodwin, Brigitte C. Widemann, Pamela L. Wolters

**Affiliations:** 1Department of Behavioral Sciences and Social Medicine, Florida State University, Tallahassee, FL 32306, USA; 2Pediatric Oncology Branch, National Cancer Institute, Bethesda, MD 20814, USA; 3Clinical Research Directorate, Frederick National Laboratory for Cancer Research, Frederick, MD 20702, USA

**Keywords:** neurofibromatosis type 1, plexiform neurofibromas, socioemotional function, behavioral problems, internalizing problems, externalizing problems, school problems, adaptive skills, personal adjustment

## Abstract

**Simple Summary:**

Individuals with neurofibromatosis type 1 (NF1) are vulnerable to socioemotional difficulties. However, we know little about how socioemotional functioning changes across childhood and adolescence and what factors predict the change. The current longitudinal study aims to address this gap by examining how demographic and NF1-disease related factors predict development of socioemotional functioning in a sample of children and adolescents with NF1 and PNs. We found substantial within-group variabilities in socioemotional development with developmental patterns varying across socioemotional domains, reporters of socioemotional functioning, age at baseline, parental education, race/ethnicity, number of NF1-related complications, and visibility of tumors. These findings indicate the need for individualized patient care and interventions that consider each child’s unique background and needs. The results also suggest the need for more longitudinal research to further understand how various factors affect socioemotional development in children with NF1 to better inform intervention development and practices.

**Abstract:**

Individuals with neurofibromatosis type 1 (NF1) and plexiform neurofibromas (PNs) have a higher risk for socioemotional problems. The current study aims to identify the socioemotional developmental pattern and its predictors across childhood and adolescence in individuals with NF1 and PNs. Participants included 88 children with NF1 and PNs (aged 6–18 years old, M = 12.05, SD = 3.62, 57% male) in a natural history study. Neuropsychological assessments were administered three times over six years. There are large variabilities in socioemotional development in the study participants. Developmental patterns varied across socioemotional domains, respondent type (parent-report [PR] vs. child-report [CR]), demographic factors, and NF1 disease-related factors. For instance, lower parental education was associated with a greater decline in internalizing problems (PR) but a greater increase in school disconnectedness (CR) over time. Non-White (vs. White) children were more likely to experience increased adaptive skills (PR) but decreased personal adjustment (CR). Children with more visible tumors experienced a greater decrease in school disconnectedness (CR). Children with more NF1 complications experienced a greater decrease in externalizing problems (PR). These findings indicate the necessity of using multi-informants and investigating subdomains of socioemotional functions. They also highlight the importance of developing individualized approaches to patient care and interventions.

## 1. Introduction

Neurofibromatosis type 1 (NF1) is a genetic disorder caused by mutations in the NF1 tumor suppressor gene (chromosome 17), affecting approximately 1:3500 individuals [1]. NF1 has multiple tumor manifestations, such as plexiform neurofibromas (PNs) and cutaneous neurofibromas, as well as non-tumor manifestations, including various physical complications (e.g., bone abnormalities) [1]. Cognitive impairments and socioemotional problems are also common in NF1 [2,3]. Most existing studies on socioemotional functioning of individuals with NF1 have used a cross-sectional design to compare individuals with NF1 to the normative population or a control group [4,5,6,7,8]. Although these studies are informative regarding the prevalence and extent of socioemotional problems, we know less about how socioemotional functions change across time within the NF1 group and what factors predict their socioemotional development. The current study aims to describe how domains of socioemotional functioning (i.e., internalizing problems, externalizing problems, adaptive skills, personal adjustment, school connectedness) change across childhood and adolescence and to identify predictors of socioemotional development in children with NF1. Identifying developmental patterns and predictors of socioemotional functioning will provide important information for patient management and inform the development and optimal timing of interventions.

### 1.1. Socioemotional Development in the General Population

In the general population, past longitudinal research on socioemotional development identified a stable or improving trend from childhood to adolescence in outcomes such as internalizing and externalizing problems [9,10], hyperactivity/impulsivity, inattention [11,12,13], school connectedness [14], and adaptive skills [15,16]. Moreover, extant studies have moved beyond describing the average developmental patterns to highlight heterogeneity in socioemotional development, demonstrating that developmental patterns could vary across socioemotional domains, developmental periods, and demographic groups (e.g., sex, socioeconomic status, race, family structure). For example, externalizing problems tend to decrease throughout childhood [17,18], increase during adolescence, and then decrease from late adolescence to adulthood [18]. Males (vs. females) were more likely to show decreasing internalizing problems [19], high stable adaptive functioning across childhood and adolescence [16], and slower development of adaptive skills [15]. Higher (vs. lower) socioeconomic status was related to better mental health [20] and school connectedness [14]. Ethnic minority groups such as African Americans (vs. White) tended to have less favorable developmental trajectories in school connectedness [14] and externalizing problems [10], although some studies showed mixed racial/ethnic differences across reporters [17]. Children from single-parent families (vs. two-parent families) were more likely to develop internalizing, externalizing, school, and social relationship problems over time [21,22,23,24].

### 1.2. Socioemotional Development in Children with NF1

An increasing number of studies have demonstrated that children with NF1 have more socioemotional problems than the normative population. For example, children with NF1 are at a greater risk for autism and ADHD symptoms [4,8,25], more internalizing symptoms, social interaction problems, and maladaptive behaviors [5,6,7,26,27], and poorer adaptive skills [6,28], compared to their unaffected siblings or the normative group. Most extant studies are cross-sectional, leaving the developmental patterns of socioemotional functioning in NF1 underexplored.

The few longitudinal studies in children with NF1 showed mixed findings on how socioemotional functions change across time. For example, one study found no significant change in the socioemotional impairment of 43 children with NF1 from infancy to preschool age [29]. Similarly, there was no significant change in risk for atypical behaviors in 39 young children with NF1 from 21 to 40 months old [30]. In contrast, the adaptive skills of 26 children with NF1 generally decreased over time from early childhood to school age [31]. Another study demonstrated that internalizing (but not externalizing) problems significantly increased across three years in a group of young children from 2 to 6 years old (*n* = 23) [32]. These mixed findings could be due to developmental differences across socioemotional domains, developmental periods, or within-group socioemotional variabilities related to various demographic and NF1 disease-related factors. More longitudinal studies are urgently needed to understand better how socioemotional functions change across childhood and adolescence within children with NF1 and to identify potential predictors accounting for within-group differences.

### 1.3. Demographic and Disease-Related Predictors of Socioemotional Development in Children with NF1

There is a lack of studies investigating predictors of socioemotional development in children with NF1, as most existing studies focused on comparing socioemotional functions between the NF1 and control groups. A few available studies showed that demographic factors such as age, sex, and socioeconomic status were not related to socioemotional problems such as internalizing problems, externalizing problems, and daily life behavioral problems related to executive functioning [33,34]. These preliminary findings in children with NF1 are in contrast with the demographic differences identified in the general population reviewed above. However, before making interpretations about children with NF1 based on these initial results from a limited number of studies with small sample sizes, future studies, preferably longitudinal investigations, are needed to understand better whether demographic differences found in the general population exist in the same extent and direction in the NF1 population.

Besides demographic factors, children with NF1 also vary in aspects of their medical condition, such as the severity, visibility, and inheritability of the disease, which may account for individual differences in their socioemotional development. A few studies on how NF1 disease severity related to socioemotional functions demonstrated mixed findings [7,33,34]. For example, one study found that the presence of specific NF1-related disease complications (e.g., lisch nodules) was not related to internalizing and externalizing problems or executive functioning-related behavioral problems [33]. In contrast, other studies demonstrated that more NF1-related disease complications were related to more internalizing problems [7], and greater disease severity was associated with less positive emotions [35]. Regarding disease visibility, limited existing studies have shown that greater perceived disease visibility was associated with more socioemotional problems such as depression, psychological stress [36], and negative emotions [35]. For NF1 inheritability, a few cross-sectional studies showed no difference in socioemotional functions between the familial and sporadic forms of NF1 [33,37,38]. However, it is still worthwhile to explore whether NF1 heritability may relate to changes in socioemotional functioning across time, as a prior study found that NF1 heritability was related to change (despite not initial levels) in some cognitive functions (e.g., working memory, attention, reading) over time [39]. Therefore, the current study examined three disease-related factors potentially related to the socioemotional development of children with NF1 and PNs, including NF1 disease severity, NF1 disease visibility, and NF1 inheritability.

### 1.4. The Current Study

The current study had two primary aims. Aim 1 was to describe the longitudinal developmental patterns of socioemotional functions in children with NF1. Aim 2 was to identify demographic and disease-related predictors that account for within-group variabilities in socioemotional development. This study focused specifically on children with NF1 and PNs, as socioemotional development can potentially differ between children with versus without PNs. PNs are one type of central nervous system tumor, existing in about 50% of children with NF1 [40]. One prior study demonstrated that among children with NF1, children with (vs. without) central nervous system tumors have more cognitive impairments and socioemotional problems (e.g., internalizing problems and oppositional-defiant behaviors [41].

The current study extended prior research in multiple ways. First, the limited existing longitudinal studies on socioemotional development in children with NF1 tend to focus on younger children. This study was the first to investigate socioemotional development across childhood to adolescence and directly tested whether developmental patterns vary across age groups. Second, this study was among the first to test how demographic and disease-related factors relate to changes in socioemotional functions across time. Third, the present study had a relatively larger sample than prior longitudinal studies, allowing greater power to detect potential predictors. Furthermore, both parent-reported and adolescent-reported measures of socioemotional functions were included to explore how results vary across socioemotional measures, given that prior studies have demonstrated inconsistent findings based on different socioemotional measures [34,42,43].

## 2. Materials and Methods

### 2.1. Participants and Inclusion Criteria

Individuals who had been diagnosed with NF1 in terms of the National Institutes of Health Consensus Conference criteria [44] or had a confirmed NF1 germline mutation with analysis performed in a CLIA-certified laboratory were eligible for a natural history study (ClinicalTrials.gov Identifier: NCT00924196) at the National Cancer Institute (NCI). The protocol was approved by the NCI Institutional Review Board. As specified in the protocol, individuals under 35 years of age enrolled in this study were eligible to undergo a comprehensive neuropsychological evaluation three times at approximately three-year intervals. Young people (ages 6 to 18 years) who completed at least one neuropsychological evaluation and were diagnosed with at least one PN were eligible for this sub-study examining socioemotional functioning. The Data cutoff for this psychological sub-study was 20 May 2019.

### 2.2. Measures

#### 2.2.1. Socioemotional Functioning

Children’s socioemotional functioning was assessed using the Behavior Assessment System for Children (BASC 2) [45,46] at three assessment times. The parent-reported (PR) BASC 2 was administered to parents or primary caregivers (referred to as parents hereafter) of children 6–18 years old. It includes three composite scores and 11 subscales: Externalizing Problems (i.e., Conduct Problems, Hyperactivity, Aggression), Internalizing Problems (i.e., Anxiety, Depression, Somatization), and Adaptive Skills (i.e., Adaptability, Social Skills, Leadership, Activities of Daily Living, Functional Communication). The self-reported (SR) BASC 2 was administered to children 8–18 years old. It includes the following composites: Inattention/Hyperactivity (i.e., Attention Problems, Hyperactivity), Internalizing Problems (i.e., Anxiety, Depression, Social Stress, Sense of Inadequacy, Somatization [for adolescent only]), School Disconnectedness (i.e., Attitude to School, Attitude to Teachers, Sensation Seeking [for adolescent only]), and Personal Adjustment (i.e., Self-Esteem, Self-Reliance, Relationships with Parents, Relationships with Peers). The current study focused on composite scores. Higher scores indicate better functioning for adaptive skills (PR) and personal adjustment (CR). For other composites, higher scores indicate worse socioemotional functioning.

#### 2.2.2. Demographic Predictors

Demographic information was reported by parents or primary caregivers at baseline, including children’s age, sex (0 = female, 1 = male), parental education, race, and family structure. Maternal and paternal education in years were reported separately and averaged to represent parental education. Participants reported whether the child’s race was White, African American, Hispanic, Asian, or others. We recoded race as White versus non-White. Family structure included two categories: families with 2 parents/guardians versus 1 parent/guardian.

#### 2.2.3. Disease-Related Predictors

Parents or primary caregivers (referred to as parent hereafter) and nurse practitioners reported disease-related factors at baseline. Parents reported whether the mother and father have NF1. NF1 inheritability was categorized as either “no parent has NF1” or “at least one parent has NF1”. NF1 disease severity was assessed by two measures: 1) the presence or absence of 17 NF1-related disease complications (e.g., scoliosis, spinal fusion, vision problems) rated by the nurse practitioner, which were summed for a total score [7]; and 2) severity of NF1 symptoms rated by parents on a three-point scale (1 = mild, 2 = moderate, and 3 = severe). The frequency for each category of severity of NF1 symptoms was mild 29 (33%), moderate 49 (56%), and severe 10 (11%). Parents also reported visibility of tumors on the same three-point scale. The frequency for each category of visibility of tumors was mild 38 (43%), moderate 40 (46%), and severe 10 (11%). We recorded these parents’ reported variables into two categories (“mild” versus “moderate or severe”) given the low frequency of severe ratings.

### 2.3. Analysis Plan

We used one-sample *t* tests (one-tailed) to compare patients’ mean scores on each measure to the normative mean. We described the percentage of children whose scores of socioemotional functioning fall into At Risk or Clinical Significance (AR/CS, ≤40 for adaptive skills and personal adjustment, ≥60 for other scales) [45,46]. Then, we adopted the same multilevel growth modeling approach used in our prior paper [39] to estimate change in socioemotional functions across time using the “nlme” package in R [47]. The multilevel modeling approach has several advantages in analyzing longitudinal data compared to traditional approaches often used in prior studies, such as repeated measures analysis of variance (ANOVA). For example, multilevel modeling (a) does not require the balanced design or equally spaced measurements that are needed by ANOVA, (b) treats time as a continuous variable instead of a categorical variable as assumed in ANOVA, and (c) allows researchers to assess different-level variations and pose hypotheses about relations occurring at each level and across levels [48].

In multilevel modeling analyses, first, we tested a set of unconditional multilevel models to calculate the intraclass correlation coefficient (ICC) for each socioemotional function. Second, we tested unconditional multilevel growth models with assessment time as the predictor to estimate the average change pattern of each socioemotional function across time. We also made model comparisons among unconditional multilevel growth models to determine which model fits the data best: the models with or without random intercept and/or slope. A better model was determined by smaller values of Akaike information criterion (AIC) and the Bayesian information criterion (BIC) [49]. Third, based on the best unconditional multilevel growth models, each of the potential demographic and disease-related predictors was tested separately in conditional growth models to examine their separate effects. Then, we simultaneously included all predictors that showed significant separate effects into the final combined models to test their unique contribution above and beyond each other. Age at baseline was included in the combined models regardless of whether it had significant separate effects or not, because to have a clear estimate of within-individual change across time, it is essential to control for between-individual differences in baseline age. When a significant interaction effect between the predictor and time was detected, we conducted simple slope tests. We plotted the interaction effect using the predicted values of socioemotional outcomes at baseline, 3 years after baseline, and 6 years after baseline at different levels of a predictor. Missing data were handled using the “na.action = na.exclude” function in R, which made use of all available data points in multilevel modeling [47]. Because there is very limited knowledge on how socioemotional functioning develops across time in children with NF1, our study is considered exploratory. Thus, alpha was set at 0.05 despite the fact that multiple models were run.

## 3. Results

### 3.1. Characteristics of Participants

Among the 176 eligible participants of the NCI natural history study, 117 participants completed at least one neuropsychological evaluation. Of these, 95 children were within the age range of 6 to 18 years when they completed at least one neuropsychological assessment; 7 participants who did not have at least one PN were excluded. Among the 88 eligible participants for the current study, the majority were White (*n* = 67, 76%; non-White group including 6 [7%] African American, 4 [5%] Hispanic, 2 [2%] Asian, 9 [10%] others), male (*n* = 50, 57%), and just under half had at least one parent with NF1 (*n* = 41, 47%). The median parental education was 14 years of school (interquartile range = 4). Parental education is higher in parents without (versus with) NF1 (*t* = 3.48, *p* < 001, Cohen’s d = 0.80). Per parent report, 55 (63%) participants received special education at some point during their development, 30 (34%) had been diagnosed with a learning disability, and 35 (40%) had been diagnosed with a psychiatric condition, including 29 (33%) with ADHD, 8 (9%) with depression, and 13 (15%) with anxiety. Twenty-four children (27%) took psychiatric medications, including 19 (22%) taking stimulant medications, 6 (7%) taking depression medication, 2 (2%) taking anxiety medication, and 7 (8%) taking sleep medication. Meanwhile, sixteen (19%) were on systemic treatment for PNs (e.g., MEK inhibitors) and 13 (15%) received therapeutic services at the time of evaluation at baseline. The average number of NF1-related disease complications (e.g., scoliosis, spinal fusion, vision problems) rated by the nurse practitioner was 4 (*SD =* 1.73). Fifty-nine participants (67%) had a moderate or severe level of NF1 symptoms rated by parents, and 50 (57%) had a moderate or severe level of visibility of tumors reported by parents.

Among the 88 participants assessed at Time 1, 65 (75%) were also assessed at Time 2, and 34 (39%) were assessed at Time 3. We compared the baseline values of all study variables between participants with and without data at Time 2 and Time 3. For example, the difference in externalizing problems (PR) at baseline was examined between 65 participants with Time 2 data and 23 without Time 2 data. Independent samples *t* tests (for continuous variables) and chi-square tests (for categorical variables) were conducted. Bonferroni correction was adopted to reduce Type I error as multiple *t* tests were conducted. Results indicated that participants with data at Time 3 (*n* = 34) were younger than participants without data at Time 3 (*n* = 54, *t* = 3.10, *p* < 0.01); participants with data at Time 2 (*n* = 65) had a higher internalizing problems score (CR) at baseline than participants without data at Time 2 (*n* = 23, *t* = −4.43, *p* < 0.001); participants with data at Time 2 (*n* = 65) had a lower level of personal adjustment (CR) at baseline than participants without data at Time 2 (*n* = 23, *t* = 4.12, *p* < 0.001).

### 3.2. Descriptive Statistics and T-Test Results of Socioemotional Outcomes

Descriptive statistics (e.g., M, SD, percentage of children with AR/CS socioemotional difficulties) and *t*-test results of all socioemotional outcomes for the three timepoints are provided in Table 1. Compared with the normative mean, parents reported significantly higher scores on child internalizing problems and lower scores on child adaptive skills at all time points. The percentage of children with AR/CS socioemotional difficulties varied across socioemotional domains and timepoints (8–52%). At baseline, the highest percentage of children with AR/CS scores was in internalizing problems (PR, 37%), followed by adaptive skills (PR, 34%), internalizing problems (CR, 25%), inattention/hyperactivity (CR, 25%), school disconnectedness (CR, 18%), externalizing problems (PR, 17%), and personal adjustment (CR, 16%).

### 3.3. Change in Socioemotional Outcomes and Predictors

In terms of unconditional multilevel models (without any predictors), intraclass correlation coefficient (ICC) for each socioemotional function was calculated, which fell in the range of 0.27~0.72, meaning that 27~72% of variances in socioemotional outcomes were attributable to between-individual differences, whereas 28~73% of variance in socioemotional outcomes were attributable to within-individual changes across time. According to multilevel growth models (with time as a predictor), model comparisons showed that models with random intercepts fit the data best, except that for adaptive skills (PR), the model with random intercept and slope fit the data best. Regarding general patterns of change, there was not a significant change in standardized scores of seven socioemotional outcomes. However, there were considerable between-individual variations of initial levels and/or change patterns across all socioemotional outcomes, as shown by ICC and Appendix A. Therefore, individual predictors could be added to the multilevel growth models to explain variance in socioemotional outcomes.

Results for separate models are presented in Appendix A. Table 2 presents the results from the combined models with all predictors that had significant separate effects, and the following is a discussion of the most robust predictors that had significant unique effects on socioemotional functioning above and beyond other predictors.

In the combined models, there was a negative interaction between age at baseline and assessment time on internalizing problems (PR, *β* = −0.34, *SE* = 0.10, *p* < 0.05). Specifically, younger children (vs. older) experienced a greater increase in internalizing problems (PR) across time (Figure 1).

Meanwhile, age at baseline was negatively associated with the initial status of school disconnectedness (CR, *β* = −0.97, *SE* = 0.35, *p* < 0.01), suggesting that younger children encountered more school disconnectedness at baseline. There was a positive interaction effect between parental education and time on internalizing problems (PR, *β* = 0.32, *SE* = 0.14, *p* < 0.05). In contrast, there was a negative interaction effect between parental education and time on school disconnectedness (CR, *β* = −0.52, *SE* = 0.20, *p* < 0.05). Lower parental education was related to a greater decrease in internalizing problems (PR) over time (Figure 2a) but a greater increase in school disconnectedness (CR) across time (Figure 2b).

Moreover, there was a negative interaction effect between children’s race and time on adaptive skills (PR, *β* = −1.46, *SE* = 0.71, *p* < 0.05) but a positive interaction effect between children’s race and time on personal adjustment (CR, *β* = 1.79, *SE* = 0.89, *p* < 0.05). Non-White children (vs. White children) exhibited a greater increase in parent-reported adaptive skills (Figure 3a) but a greater decrease in self-reported personal adjustment (Figure 3b).

Children from single-parent families (vs. two parent families) had a higher level of personal adjustment (CR, *β* = 4.32, *SE* = 2.09, *p* < 0.05) at baseline. Additionally, there was a negative interaction effect between parent-reported visibility of tumors and time on school disconnectedness (CR, *β* = −1.72, *SE* = 0.75, *p* < 0.05). Specifically, children with a moderate or severe (vs. mild) level of tumor visibility experienced a greater decline in school disconnectedness (Figure 4).

Parent-rated severity of NF1 symptoms was positively related to the initial status of internalizing problems (PR, *β* = 7.46, *SE* = 2.58, *p* < 0.01). There was a negative interaction effect between the number of NF1-related disease complications rated by the nurse practitioner and time on externalizing problems (PR, *β* = −0.28, *SE* = 0.13, *p* < 0.05). More NF1-related disease complications were related to a greater decrease in externalizing problems across time (Figure 5).

## 4. Discussion

The current study is among the first to examine longitudinal socioemotional development and its predictors across childhood and adolescence in children with NF1 and PNs. Notably, participants demonstrated more difficulties with internalizing problems and adaptive skills compared with the normative sample; group differences were not observed in other socioemotional domains. Importantly, although no mean level changes were found across the socioemotional domains, considerable within-group variability in developmental patterns existed. The developmental patterns varied across socioemotional domains, reporters of socioemotional functioning, demographic factors, and NF1 disease-related factors. Based on child-reported socioemotional functioning, having with lower parental education, being an ethnic minority (vs. White), or having less visible tumors had greater risks for less favorable socioemotional development over time, although these findings were only applicable to the domains of school disconnectedness and personal adjustment. However, based on parent-reported socioemotional functioning, child participants who were younger at baseline, White, with higher parental education, or had less NF1 complications were at greater risks for worsening socioemotional functioning over time. These findings will not only enhance our understanding of the socioemotional development in children with NF1 and PNs and potential predictors, but they also have significant implications for future research and practices/interventions. More detailed discussions of these findings are presented below.

Overall, the study results suggest that socioemotional difficulties appear to be less prominent in children with NF1 than expected from the prior literature, except in selected domains, including internalizing problems and difficulties with adaptive skills (e.g., [5,27,28]). There was no significant mean-level difference between children with NF1 and the normative group in most socioemotional domains examined, including externalizing problems, school disconnectedness, inattention/hyperactivity, and personal adjustment. The percentages of children with NF1 that reached clinical significance in these socioemotional subdomains (8% to 25%) were also similar to what is expected in the general population (about 16%) [45,46]. However, child participants demonstrated significantly higher levels of internalizing problems, which included anxiety, depression, and somatization, as well as lower adaptive skills than the normative means across times based on parent report. The proportion of the study sample showing clinically significant internalizing problems (32% to 37% across time) and difficulties with adaptive skills (34% to 52% across time) was two to three times higher than that expected in the general population. Impairments in these two socioemotional domains also were found in previous studies [5,26,28], reaffirming high risks in children with NF1 of internalizing problems and adaptive skills deficits. There also were some fluctuations in mean levels and proportions of participants showing clinically significant impairment across time. For example, the mean level of personal adjustment in the study sample was lower than the normative mean at Time 2 but not at Times 1 and 3, with a high percentage of participants showing clinically significant impairment at Time 2 but low percentages at Times 1 and 3. Taken together, these results suggest that socioemotional functions in children with NF1 vary across socioemotional domains, reporters, and developmental periods, although internalizing problems and adaptive skills appear to be domains at particular risk. Findings highlight the importance of continuously monitoring various socioemotional functions and utilizing multi-informants to collect information.

No significant mean level changes in the examined socioemotional functions were found, which inconsistent with findings in the general population who tend to experience improved socioemotional development from childhood to adolescence (e.g., [10,13,14,15]). This lack of improvement might be because children with NF1 face more physical challenges [1,2,3] or are less adaptive in general, although individual differences are also common. Impairments in cognitive abilities frequently found in the NF1 population [2,3] may have also contributed to the continuing socioemotional difficulties [6,50]. Indeed, improvements in development over time are rarely found among children with NF1 [29,30], and some previous research has found worsening problems, including increasing internalizing problems and poor adaptive skills over time [31,32]. The current study also found persistently poorer adaptive skills in the sample than in the normative sample over time. Thus, there exists a great need for long-term interventions that target these domains starting at an early age.

More importantly, considerable within-group variabilities in the development of the various socioemotional subdomains were observed in children with NF1, similar to findings in the general population (e.g., [14,16]) as well as previous studies on children with NF1 (e.g., [29]). Demographic and disease-related variables can partly account for the variabilities. First, although the mean scores of all tested socioemotional domains were in the normal range (>40 for adaptive skills and personal adjustment, <60 for others) [45,46] for children of different ages, children who were younger at baseline appeared to be at greater risks for poorer functioning in certain socioemotional domains including internalizing problems and school disconnectedness. Specifically, younger children at baseline were more likely to experience increased internalizing problems over time. This finding is not surprising, as research on the general population suggests that internalizing problems increase steadily from childhood to adolescence, with younger children showing more rapid increases, which gradually level off during adolescence [51]. Younger children also reported higher levels of school disconnectedness at baseline, consistent with a previous finding based on the general population that children may experience improved school connectedness as they grow older [52]. Children’s sex was not related to the initial levels or changes in any of the socioemotional domains. Although inconsistent with research on the general population [15,16,19], the finding is consistent with previous research on children with NF1 [34], suggesting that males and females with NF1 and PNs are likely to experience socioemotional development similarly.

Effects of parental education and race showed mixed findings across reports of socioemotional functions. For child-reported socioemotional functions, those with a higher (vs. lower) parental education experienced a greater decline in school disconnectedness (i.e., improved school connectedness) over time, possibly because parents with higher education can better help with children’s school problems. Non-White or racial minority children showed a decreasing tendency in levels of personal adjustment, contrary to White children, who exhibited an improving trend in personal adjustment over time. This finding is consistent with research on the general population focusing on ethnic differences in general socioemotional difficulties and health-related quality of life [53,54]. However, based on parent-reported socioemotional functions, children with lower (vs. higher) parental education experienced a greater decline in internalizing problems over time, contrary to previous findings based on the general population [20]. In addition, non-White (vs. White) parents were more likely to perceive improving adaptive skills in children across time. Parents with lower education or from an ethnic minority group may perceive better child socioemotional functioning over time because they adjusted their expectations and became more accepting of their children with NF1. If these parents were more likely to have NF1, this may have also contributed to them being more accepting of their children with NF1. However, future studies need to investigate further what leads to such discrepancies in this particular population. Finally, children from single-parent families reported higher levels of personal adjustment at baseline than those from two-parent families, inconsistent with findings from the general population [21,22,23,24]. This finding might be related to potential unexamined protective factors for these children, such as social support, parent internal strengths (e.g., optimism and self-efficacy), and parenting practices, as shown in research on single-parent families in the general population [55]. Future research is needed to explore this possibility.

NF1 disease-related factors also were associated with children’s socioemotional development. More severe NF1 symptoms were associated with a higher initial level of internalizing problems, consistent with findings in previous cross-sectional studies [7,35,36]. The current study moved beyond prior studies to explore how NF1 disease-related factors were related to changes in children’s socioemotional functions. It was found that a higher number of NF1-related disease complications at baseline was related to a greater decrease in externalizing problems across time. It appears that children with more NF1 complications at baseline had more rapid improvement in externalizing problems over time. This result may be because as children with NF1 get older, their ways of coping shift away from using more externalizing behaviors to other behaviors, possibly internalizing, as it was found to be one of the most affected domains in study participants across all assessments. Similarly, children with more visible tumors reported increased school connectedness across time, suggesting that children who had more visible tumors were initially able to develop better connections with the school over time. It is possible that individuals with more visible tumors of NF1 may cope better in some respects, such as feeling more connected at school, because people can see their difficulties and have empathy. However, individuals with less visible tumors might withdraw from peers or activities and feel less connected because other people are less supportive due to not seeing and understanding the medical difficulties they are coping with. There are several additional plausible explanations for these seemingly counterintuitive findings. First, children with mild NF1 symptoms are likely to experience increased complications or grow more visible tumors over time, possibly accompanied by increased or slower improvement in socioemotional impairment. Second, although children with more severe NF1 symptoms might experience high levels of difficulties with socioemotional function initially, they may be able to better accept and, thus, better adjust to the disease over time. Moreover, certain unexamined factors such as resilience and positive family relationships [35] or parenting may have contributed to the positive adjustment observed in children with more severe NF1 symptoms. Future research should try to replicate the findings and further investigate the potential mechanisms explaining the results.

### 4.1. Implications

The current study adds to the growing literature documenting socioemotional development in children with NF1. Although development in some socioemotional domains appeared to be comparable to the normative sample, there were greater levels of internalizing and adaptive problems in children with NF1 and PNs. This finding suggests that it would be valuable to develop intervention programs that target these socioemotional domains. For example, acceptance and commitment therapy (ACT) might be implemented to improve their socioemotional functioning, as a pilot study has shown its effectiveness in helping individuals with NF1 and PNs cope with their chronic pain [56], as well as the potential to improve socioemotional functioning such as pain-related anxiety [57]. Study results also highlight within-group variabilities in socioemotional development which varied across children’s age, parents’ education, race/ethnicity, and family structure, as well NF1 disease severity (i.e., number of complications, degree of symptom severity, and visibility of tumors). These results indicate the need to provide individualized patient care and interventions that consider each child’s unique background and needs.

The current study also has important implications for future research. It is important for researchers to utilize multiple informants, as the present study showed different findings based on parent and child reports of socioemotional functioning. It would be interesting to test how changes in NF1 physical symptoms and cognitive skills are associated with changes in socioemotional functioning across time. To achieve this, future studies need to include a larger longitudinal sample with multiple assessments of these relevant constructs. Lastly, study findings showed more positive development in some socioemotional domains in individuals that were expected to have less favorable development over time based on the extant literature, such as children with lower parental education, children who belong to a minority racial/ethnic group or those from a single-parent family, and children with a higher number of NF1-related complications and more visible tumors. More longitudinal studies are needed to test the replicability of these results and to further investigate factors that may contribute to positive development, such as child resilience, peer relationships, and family functioning. These investigations will provide new directions for interventions/practices to help improve the socioemotional development of children with NF1.

### 4.2. Limitations

There are limitations to our study that should be considered when interpreting the results. First, the current study focused on a sub-population of NF1 patients with PNs, who make up about half of the population with NF1 [40]. Thus, the generalizability of the study findings to other children with NF1 needs to be tested. Second, the sample size for the current study is somewhat small, albeit larger than most previous longitudinal studies [29,30,31,32]. A larger longitudinal sample will provide more robust tests of potential predictors and permit investigation of more complicated questions such as how cognitive and socioemotional functions co-develop across time. Given the challenges of collecting a large longitudinal sample of this rare disease for any individual site, collaborative work that involves multiple sites is highly recommended. Third, considerable attrition occurred throughout the three assessment timepoints. We encourage the conduct of future studies with a larger longitudinal sample and less attrition to test the replicability of the findings. That said, the “na.action = na.exclude” function in R applied with long-format data in multilevel modeling made use of all available data points, minimizing the impact of attrition. Fourth, the current study did not include a control group, and thus the findings were compared to the normative sample, general population, and prior NF1 studies. Future research with a sibling control group may better show how NF1 disease-related factors link to socioemotional development as siblings share similar socioeconomic status and family environments. Lastly, many other factors not analyzed in the current study may also influence socioemotional development in children with NF1, for example, NF1-related symptoms, neurological deficits, cosmetic disfigurement, and cognitive functioning. We recommend future studies to explore how these additional factors relate to socioemotional development in children with NF1.

## 5. Conclusions

The current investigation is among the first to utilize longitudinal data spanning from childhood to adolescence to study socioemotional development in children with NF1 and PNs. Compared to the normative sample, children with NF1 and PNs demonstrated more internalizing problems and poorer adaptive skills but were similar in other socioemotional domains. Findings highlight substantial individual variabilities in socioemotional development, with some individuals showing improving development while others showing worsening development over time. We identified several demographic and NF1 disease-related factors that can partially explain the individual differences in socioemotional development, including age, parental education, race/ethnicity, family structure, number of NF1-related complications, and visibility of tumors. The results suggest the need for individualized patient care and interventions. Additional longitudinal research is needed to replicate our findings and to understand further how various factors affect socioemotional development in children with NF1. This will better inform intervention development with the goal of aiding healthy development and quality of life in children with NF1.

## Figures and Tables

**Figure 1 cancers-14-05956-f001:**
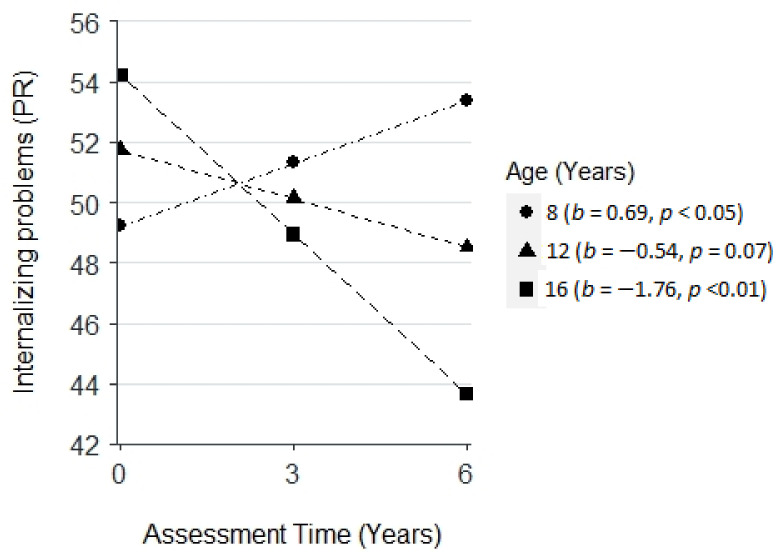
Plot for the interaction effect among assessment time and age at baseline on internalizing problems (parent-reported, PR). Three levels of baseline age were plotted: 8 years (1 SD below the mean), 12 years (the mean), and 16 years (1 SD above the mean).

**Figure 2 cancers-14-05956-f002:**
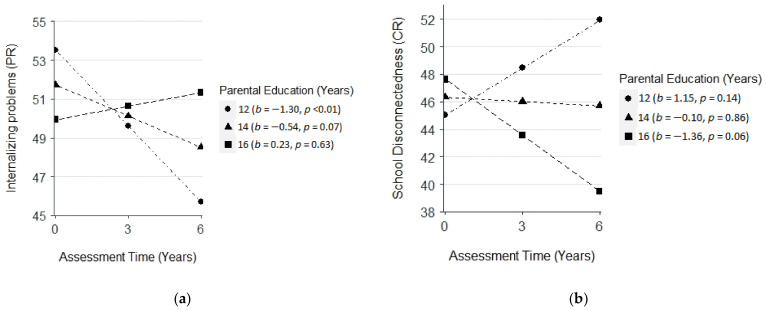
(**a**) plot for the interaction effect between assessment time and parental education on internalizing problems (parent-reported, PR); (**b**) plot for the interaction effect between assessment time and parental education on school disconnectedness (child-reported, CR). Three levels of parental education were plotted: 12 years (1 SD below the mean), 14 years (the mean), and 16 years (1 SD above the mean).

**Figure 3 cancers-14-05956-f003:**
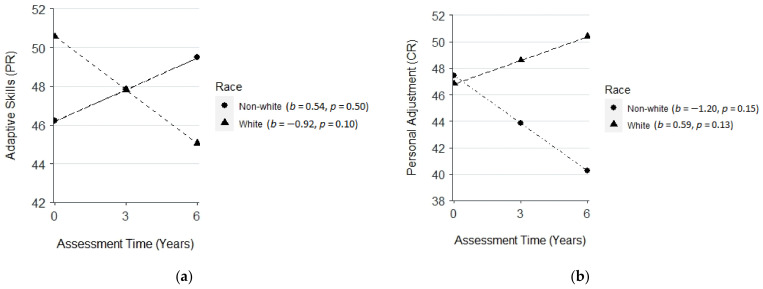
(**a**) plot for the interaction effect among assessment time and race on adaptive skills (parent-reported, PR); (**b**) plot for the interaction effect among assessment time and race on personal adjustment (child-reported, CR).

**Figure 4 cancers-14-05956-f004:**
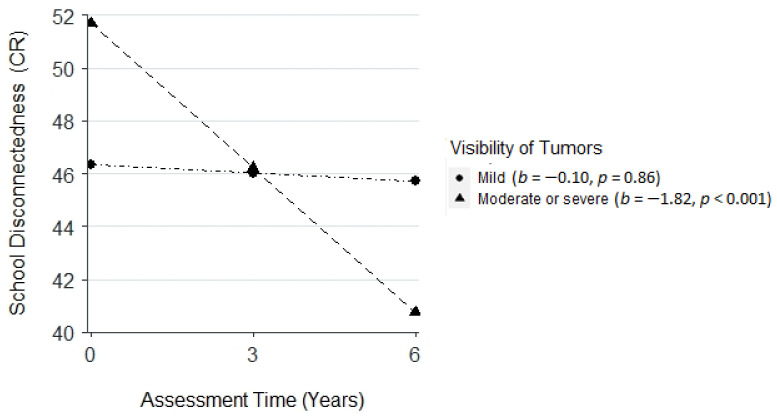
Plot for the interaction effect among assessment time and visibility of tumors on school disconnectedness (child-reported, CR).

**Figure 5 cancers-14-05956-f005:**
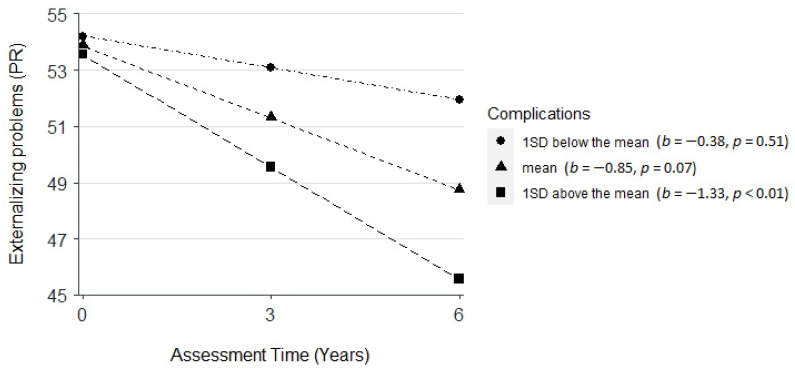
Plot for the interaction effect among assessment time and NF1-related disease complications on externalizing problems (parent-reported, PR).

**Table 1 cancers-14-05956-t001:** Descriptive Statistics and *T*-Test Results for Socioemotional Outcomes and Age at Each Assessment Time.

	Baseline					3 Years after Baseline					6 Years after Baseline				
	*n*	Mean	*SD*	*t ^a^*	% AR/CS	*n*	Mean	*SD*	*t ^a^*	% AR/CS	*n*	Mean	*SD*	*t ^a^*	% AR/CS
Externalizing Problems (PR)	86	51.57	9.85	1.48	17	56	50.79	8.80	0.67	16	25	49.64	10.60	−0.17	16
Internalizing Problems (PR)	86	56.85	12.24	5.19 ***	37	56	56.45	11.67	4.13 ***	36	25	55.16	11.13	2.32 *	32
Adaptive Skills (PR)	86	45.59	9.66	−4.23 ***	34	56	46.41	10.33	−2.60 ***	34	25	45.24	11.23	−2.12 *	52
School Disconnectedness (CR)	67	48.87	11.43	−0.81	18	52	46.83	10.84	−2.11	19	23	46.61	11.73	−1.39	13
Internalizing Problems (CR)	66	49.89	9.22	−0.09	25	55	49.51	11.08	−0.33	18	25	47.12	7.68	−1.88	20
Inattention/Hyperactivity (CR)	67	51.51	11.95	1.03	25	55	51.44	9.99	1.07	18	25	51.64	10.59	0.77	20
Personal Adjustment (CR)	67	48.96	9.47	−0.90	16	54	47.15	10.85	−1.93 *	30	25	51.00	9.66	0.52	8
Age	88	12.05	3.62			65	15.08	3.63			34	16.86	2.71		

Note: PR = parent-reported; CR = child-reported. ***^a^***
*T*-test results comparing T-scores of study sample to the normative mean of 50 (*SD* = 10). AR/CS = At-Risk/Clinically Significant range (≤40 for adaptive skills and personal adjustment, ≥60 for others). * *p* < 0.05; *** *p* < 0.001.

**Table 2 cancers-14-05956-t002:** Multilevel Modeling Coefficients of Predictors of Socioemotional Outcomes in the Combined Models.

	Externalizing Problems (PR)		Internalizing Problems (PR)		Adaptive Skills (PR)		School Disconnectedness (CR)		Personal Adjustment (CR)	
Parameters	*β*	*SE*	*β*	*SE*	*β*	*SE*	*β*	*SE*	*β*	*SE*
Intercept	53.88 ***	2.25	51.73 ***	2.10	46.21 ***	2.52	46.34 ***	1.95	47.43 ***	2.53
Time	−0.85	0.46	−0.54	0.29	0.54	0.80	−0.10	0.57	−1.20	0.82
Age	−0.50	0.29	0.69 *	0.33	0.37	0.28	−0.97 **	0.35	−0.04	0.31
Male	−2.14	2.13								
Pedu	−0.20	0.43	−0.74	0.49	0.72	0.41	0.53	0.54		
White	−1.74	2.41			4.34	2.3			−0.61	2.71
SINGLP									4.32 *	2.09
PNF1										
Visi					−2.64	2.38	5.35 *	2.55		
Seve			7.64 **	2.58	−3.36	2.53				
Comp	−0.19	0.61	0.48	0.70						
Time × Age			−0.34 **	0.10						
Time × Male	0.35	0.41								
Time × Pedu	0.19	0.10	0.32 *	0.14	−0.22	0.16	−0.52 *	0.20		
Time × White	0.58	0.50			−1.46 *	0.71			1.79 *	0.89
Time × SINGLP										
Time × PNF1										
Time × Visi					1.25	0.72	−1.72 *	0.75		
Time × Seve					0.22	0.78				
Time × Comp	−0.28 *	0.13								

Note: *β* = regression coefficients in multilevel growth models; *SE* = standard error; Age = age at baseline; Pedu = parental education; SINGLP = single parent; NF1= neurofibromatosis type 1; PNF1 = parental NF1 status; Visi = visibility of tumors; Seve = severity of NF1 symptoms; Comp = NF1-related disease complications; PR = parent-reported; CR = child-reported. Valid number of participants (*n*) and valid number of observations (*k*) for the five models were: externalizing problems (PR), *n* = 86, *k* = 167; internalizing problems (PR), *n* = 86, *k* = 167; adaptive skills (PR), *n* = 86, *k* = *167*; school disconnectedness (CR), *n* = 81, *k* = 142; personal adjustment (CR), *n* = 82, *k* = 146. * *p* ≤ 0.05; ** *p* < 0.01; *** *p* < 0.001.

## Data Availability

The data presented in this study are available on request from the corresponding author. The data are not publicly available because they are still being analyzed and published.

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
