# Peer review of "Demographic and Disease-Related Predictors of Socioemotional Development in Children with Neurofibromatosis Type 1 and Plexiform Neurofibromas: An Exploratory Study"

_cancers, 2022, doi:10.3390/cancers14235956_

Round 1
Reviewer 1 Report
This study fill the gap of the long-needed socioemotional study of NF1-associated PNs. Although the limited patient number is a concern, it is still a well-designed and summarized study.
Reviewer 2 Report
The authors should be lauded for the great effort to advance the understanding of socioemotional functioning and developmental patterns longitudinally from childhood through adolescence in a rare disease cohort such as NF1. The manuscript is very well written with a clear and concise introduction including limitations of prior relevant literature, statement of objectives, discussion of the scientific method followed by a well-rounded and comprehensive discussion of results. Analytical methods are well described. Multiple informants were used including both parent report and child report. Most results conform or validate prior reports in NF1 patient cohorts such as the higher occurrence of internalizing problems (depression, anxiety etc.) and poorer adaptive skills. Importantly, however, the longitudinal follow up demonstrates less improvement and perhaps worsening over time in these ‘at risk’ socio-emotional domains (internalizing problems and adaptive skills) as compared to a ‘normal’ comparator group, thereby indicating the need for long-term interventions that target these domains starting at an early age. There are also some interesting results that appear at first glance to be surprising/unexpected such as the effects of parental education on the internalizing problems and potential rational explanations are appropriately discussed, albeit speculative. Nonetheless these results provide a baseline on which clinical trials for identified ‘at risk’ patients may be enrolled on interventional trials. Some pertinent limitations include -
1. The lack of an appropriate comparator group and the decision to compare results from an NF1 patient cohort to a normal group instead of prior NF1 study cohorts requires more discussion/explanation.
2. The study analysis is limited to a cohort of NF1 patients enrolled on the NCI natural history study with known plexiform neurofibromas. Additional information providing details on the patient level morbidity (severity of pain and neurological deficits) or cosmetic disfigurement resulting from the PN is not provided. Might such factors influence the socio-emotional development of afflicted patients. Additional clarification in this regard would be helpful.
3. Given a third of patients had been diagnosed with a learning disability, how was this confounding factor controlled for during this analysis.
4. What was the impact on the patterns of socio-emotional development in the proportion of patients treated with MEK inhibitors? While this would be limited, it would be interesting to ascertain the impact of treatment on patients.
5. While the implications of differing outcomes between ethnic minority groups and white patients are important and highlight disparate outcomes between groups, the sample size is quite limited. Nonetheless, this represents an important observation.
6. The number of patients lost to follow-up between timepoints 1 and 2 as well as 2 and 3 is significant and limits the sample size.
Reviewer 3 Report
This manuscript nicely builds on the authors’ earlier published paper in 2020 that reported on predictors of cognitive development for the same cohort.
The authors are to be commended for a clear and well-constructed manuscript. The introduction provides a sound justification for the study and research questions, highlighting the lack of longitudinal studies investigating the socioemotional development of children with NF1 and the novelty of the current study.
The analysis was well-performed with the most robust predictors reported in the manuscript, with all results presented in the supplementary table. The inclusion of figures assisted in the interpretation of the results.
The discussion is comprehensive and well-written. Results are compared and interpreted in the context of existing NF1 studies and the general population, highlighting the significance of study findings. Implications and directions for future research are well-addressed.
Please see additional comments below.
Page 1 line 16 please add PNs to end of the sentence to clarify the sample is children and adolescents with NF1 and PNs.
Page 3 line 137 insert ‘PNs’ so it reads “children with NF1 and PNs.”
Page 5 line 210 – please provide the frequency for each category (mild, moderate, severe) for parent reported visibility of tumors.
Page 5 line 237-238 please provide further justification as to why age was included in the combined models even when it did not have a significant effect.
Page 5 line 241 please clarify that Time 1 is baseline.
Page 5 line 242 missing data was reported to be handled listwise, where a case is excluded if it has a missing value for any time point. If this is correct, please clarify the size for each analysis. The n = 25 available at Time 3 in Table 1 for measures would suggest this was the sample size over time. Except for Table 1, in all other tables and figures the sample size is not reported. It is recommended that the sample size be reported to assist in ease of interpretation. Also, missing data does not appear to be missing completely at random (MCAR) with an increasing number of cases dropping out over time and younger participants at Time 3. When data are not MCAR, listwise deletion may cause bias in the estimates of the parameters. Could the authors comment on this.
Page 6 line 256 it is stated that parental education is higher in parents with NF1 versus those without NF1. This is contrary to that reported in their Hou et al., 2020 paper that reports on the same cohort. Could the authors please clarify.
Page 6 line 283 t values are inferential statistics, not descriptive.
Page 8 Figures 1 and 8S ages are plotted for 8, 12 and 16 years for child measures. For Figures 2 and 10S, parental education is plotted at 12, 14 and 16 years. It would be valuable for the authors to explain how these ages were selected.
Pages 10 lines 341 – 349 please check all beta and SE values reported here as some do not match those reported in Table 1S. For example, in table 1S, the beta and SE value reported for parental education and internalizing problems (PR) is 0.40 and 0.15, respectively.
Page 12 line 421 It would be valuable to add here that this was only for the domains of school disconnectedness and personal adjustment.
Page 12 line 437 – please include a reference for the 16%.
Page 13 lines 468-483 while younger children were at greater risk, it is important to discuss this in the context that mean scores were in the normal range.
Page 14 lines 498 – 500 if parents with lower education were more likely to have NF1, this may have also contributed to them being more accepting of their child with NF1.
